# Conventional and Atypical Deep Penetrating Nevus, Deep Penetrating Nevus-like Melanoma, and Related Variants

**DOI:** 10.3390/biology11030460

**Published:** 2022-03-17

**Authors:** Pavandeep Gill, Phyu P. Aung

**Affiliations:** 1Department of Pathology and Laboratory Medicine, University of British Columbia, Vancouver, BC V6T 1Z7, Canada; pavandeep.gill@islandhealth.ca; 2Department of Laboratory Medicine, Vancouver Island Health Authority, Victoria, BC V8R 1J8, Canada; 3Department of Pathology, The University of Texas MD Anderson Cancer Center, Houston, TX 77030, USA

**Keywords:** deep penetrating nevus, melanocytic lesion, nevi, melanoma, metastatic, molecular

## Abstract

**Simple Summary:**

Atypical deep penetrating nevus (DPN) is a unique skin tumor with an uncertain biologic/metastatic potential that may be difficult to distinguish from DPN (an indolent lesion) and DPN-like melanoma (an aggressive lesion) based on the results of histomorphologic analysis and commonly employed molecular studies such as fluorescence in situ hybridization and comparative genomic hybridization alone. Herein, we review the clinical, histomorphological, immunohistochemical, molecular, and cytogenetic characteristics of the DPN spectrum of lesions to try to better understand the prognosis of these lesions and possible treatment approaches.

**Abstract:**

Deep penetrating nevus (DPN) is an uncommon acquired melanocytic lesion with a distinct histopathological appearance that typically behaves in an indolent manner. The lesion is characterized by a symmetrical proliferation of epithelioid to spindled melanocytes associated with abundant melanophages and wedge-shaped extension to the deep reticular dermis and subcutis. Pronounced cytologic atypia and mitotic figures are usually absent, which helps distinguish DPN from melanoma with a deep penetrating growth pattern. Recently, the concept of atypical DPN has been proposed for lesions that demonstrate borderline histomorphologic features and may be associated with lymph node deposits but lack the copy number aberrations typical of melanoma by either fluorescence in situ hybridization or comparative genomic hybridization. While most of these lesions have a favorable clinical course, rare lesions may progress to melanoma. In this review, we summarize the current literature on atypical DPNs with uncertain behavior/metastatic potential and outline the characteristics that distinguish these lesions from conventional DPN and melanoma with DPN-like features.

## 1. Introduction

Melanocytic tumors are a heterogeneous group of lesions with wide variation in histomorphology and biologic behavior. Melanocytic tumors range from conventional acquired nevi, with unequivocally bland histopathological features and completely indolent clinical courses, to melanomas, with strikingly atypical histopathological features and inarguable potential for metastatic and aggressive disease [1]. In between is a spectrum of lesions with ambiguous histopathological features and unclear biologic potential that pose challenges in regard to their correct classification and management. The most recent edition of the World Health Organization Classification of Skin Tumors outlines a series of terms that have been used to describe lesions within this category, including “superficial atypical melanocytic proliferation of uncertain significance (SAMPUS)”, “melanocytic tumor of uncertain malignant potential (MELTUMP)”, “intermediate lesion”, “melanocytic neoplasm of low malignant potential”, and “melanocytoma” [1]. Melanocytoma has been defined as “a tumorigenic neoplasm of melanocytes that generally has increased cellularity and/or atypia (compared with a common nevus) and an increased (although generally still low) probability of neoplastic progression” [1]. These borderline melanocytic lesions often pose significant diagnostic difficulty and necessitate expert consultation and ancillary molecular testing [2,3,4].

Deep penetrating nevus (DPN) is an uncommon acquired melanocytic lesion that was first reported by Seab et al. in 1989 [5]. It lies along a histological continuum with plexiform spindle cell nevus (PLEXSCN), described by Barnhill et al. in 1991 [6], and clonal/inverted type A nevus, described by Ball and Golitz in 1994 [7]. DPN has a distinct histopathological appearance and has been generally accepted as behaving in an indolent manner. It can be distinguished from melanoma with a deep penetrating growth pattern, which usually shows unequivocally aggressive histomorphological features, including pronounced cytologic atypia and significantly increased mitotic activity.

Recently, a borderline lesion has been described within the DPN spectrum of tumors, referred to as atypical DPN or deep penetrating melanocytoma. This is a histologically ambiguous lesion that demonstrates borderline features similar to those of melanoma and is sometimes associated with lymph node deposits but often lacks the copy number aberrations typical of melanoma by either fluorescence in situ hybridization (FISH) or comparative genomic hybridization (CGH) [8]. Although this lesion appears to have an overall favorable clinical course, it can exhibit potential for progression to melanoma [8,9].

In this review, we will outline the current literature on conventional DPN, atypical DPN, and DPN-like melanoma; describe the clinical, histopathological, immunohistochemical, molecular, and cytogenetic characteristics that distinguish these lesions from each other and the other DPN histological variants; and highlight treatment approaches. Atypical DPNs are an area of ongoing research, and our review describes the most current literature on this topic.

## 2. Conventional DPN

### 2.1. Clinical Features

Conventional DPN is a benign melanocytic lesion that affects patients of all ages, has a predilection for women under the age of 40 years, and is usually found on the head and neck, trunk, and upper extremities [5,10,11,12]. DPN presents as a solitary, less than 1 cm in diameter, symmetrical, well-circumscribed, nonulcerated, dome-shaped papule or nodule with dark pigmentation (Figure 1) [5,10,13]. DPN can be brown to blue to black with occasional color variegation and may seem to suddenly appear, which may raise clinical concern regarding melanoma and hence the need for biopsy [10,14]. Most patients with DPN do not have a personal or family history of melanoma [14]. DPN arising in a congenital nevus has been reported [15]. Multiple DPNs arising in linear arrangement have also been reported [16]. The dermoscopic features of DPN are not well established [14].

### 2.2. Histopathological Criteria

On histopathological examination (Figure 2), DPN is usually a small, well-circumscribed, symmetrical lesion composed of enlarged fusiform, spindled, and/or epithelioid melanocytes arranged in fascicles, cords, and nests extending into the deep dermis and subcutis in a wedge-shaped (inverted triangle) pattern [5,10,11,17]. A grenz zone may be present [13,14]. There may be peripheral collagen trapping, as seen in dermatofibromas. The bundles of melanocytes are usually associated with neurovascular and adnexal structures, giving a plexiform appearance [5,10,17]. There is no associated necrosis or loss or invasion of adnexal structures [13]. Sparse to abundant heavily pigmented melanophages and a lymphocytic infiltrate may be present [5,11,13]. There may be a limited junctional component [5,13].

Cytomorphologically, lesional cells demonstrate mild random cytologic atypia, including slight pleomorphism, variation in size, mildly to moderately enlarged nuclei, small conspicuous nucleoli, mild hyperchromatism, and occasional pseudo-nuclear inclusions [5,10,14]. Lesional cells contain a variable amount of finely granular cytoplasmic melanin [11]. Mitotic figures are rarely present (0 to 1.2 mitotic figures per square millimeter), although atypical forms are not identified [10,13,14]. There is no apparent maturation with depth [13].

DPN may exist in a pure form but is often associated with a second nevoid subtype (conventional, blue, or Spitz nevus) as part of a combined nevus (Figure 3) [10,11,12,18].

### 2.3. Immunohistochemical Features

Lesional cells in DPN are diffusely positive for S100 protein, SOX10, HMB45 antigen, melan-A/MART1, tyrosinase, and MiTF [5,19,20]. DPNs demonstrate nuclear, cytoplasmic, and membranous β-catenin expression [12,21,22], nuclear and cytoplasmic cyclin D1 expression [12,22], and nuclear lymphoid enhancer-binding factor 1 (LEF1) expression [23]. Combined nevi with DPN may show only nuclear β-catenin expression within the DPN component [24]. Immunohistochemical studies assessing *BRAF* V600E can be positive, more often in DPN that is part of a combined nevus than in pure DPN [12]. The proliferating cell nuclear antigen proliferative index is low (<5%) in DPNs, whereas it is high in nodular melanomas [25]. DPNs have been shown to retain immunostaining for dipeptidyl peptidase IV, which is lost in nodular melanomas [26]. DPNs have been shown to lack immunostaining for ataxia telangiectasia-mutated protein, in contrast with the immunoreactivity in nodular melanomas [27]. DPNs usually do not show expression for preferentially expressed antigen in melanoma (PRAME), although this is an area of ongoing investigation [28,29].

Some immunohistochemical features of DPNs are highlighted in Figure 4.

### 2.4. Molecular Biology

DPNs demonstrate activation of the WNT pathway, most commonly through gain-of-function mutations of *CTNNB1* (exon 3), which encodes β-catenin protein, and rarely through inactivation of *APC* [22]. WNT signaling increases melanocyte size and pigmentation, which results in the apparent lack of maturation seen in DPNs [22]. DPNs show diffuse expression of *AXIN2*, a marker of β-catenin transcription, by RNA in situ hybridization [22]. Mutations of genes in the mitogen-activated protein kinase (MAPK) pathway, such as *BRAF* (usually p.V600E), *MAP2K1* (also referred to as *MEK1*), or *HRAS*, may also be present [22]. Most DPNs arise from common acquired nevi, in which case the *CTNNB1* mutation is limited to only the DPN component, whereas the MAPK-activating mutation is found in both components [12,22]. Mutations of the β-catenin and MAPK pathways result in activation of LEF1, a transcription factor that facilitates the epithelial-mesenchymal transition and promotes tumorigenesis [23].

Although DPNs were once thought to be closely related to blue nevi given their shared diffuse expression of HMB45, *GNAQ* and *GNA11* mutations, which are common in blue nevi, are not found in DPNs [30]. However, the presence of *HRAS* mutations in DPNs does suggest a possible relationship with Spitz nevi [30]. *ALK* rearrangements are not a common feature in DPNs [31].

### 2.5. Cytogenetic Findings

Conventional DPNs lack cytogenetic abnormalities on FISH or CGH.

### 2.6. Prognosis and Treatment

Conventional DPNs are benign, rarely progress to melanoma, and typically are not associated with local recurrences and distant metastases [5,14,32]. Treatment consists of conservative and complete surgical excision [13,14].

## 3. Atypical DPN

### 3.1. Clinical Features

Atypical DPNs (also known as deep penetrating melanocytomas, atypical DPNs with uncertain malignant potential, and borderline DPNs) are considered intermediate-grade lesions and are an area of ongoing study with limited literature. These are rare tumors that have more atypical clinical features than conventional DPNs, including larger size and more frequent asymmetry [1]. An early review of seven cases by Magro et al. showed a male predilection and mean age at presentation of 22.3 years (range, 14–36 years) [9]. In a subsequent review of 40 atypical DPN–type lesions, Magro et al. showed that these lesions had a slight female predilection (1.5:1 female to male ratio) and wide range of age at presentation (range, 10–62 years), with a median age at presentation of 34.5 years [8]. The lesions were more commonly distributed on the face, upper and mid back, and forearm and rarely on the lower torso and lower extremity [8]. A series of 13 cases reported by Muhlbauer et al. also confirmed a female predilection and showed a mean age of 36 years (range, 11–66 years) and the back and arm as common locations [3]. In a series of 21 cases, Manca et al. showed no sex or anatomic predilection, median age of 27 years (range, 15.5–45 years), and mean diameter greater than 5 mm [33]. Abraham et al. reported a single case of an atypical DPN on the face of a 4-year-old boy [34]. Isales et al. reported a single case of an atypical DPN on the thigh of a 53-year-old woman [35].

### 3.2. Histopathological Criteria

Atypical DPN may arise in a background characteristic of conventional DPN. Compared to conventional DPN, atypical DPN has increased architectural and cytologic atypia. However, the atypical features fall short of those diagnostic of melanoma (Figure 5). Atypical architectural features include asymmetry, hypercellularity with expansile nodular or diffuse sheet-like architecture, and infiltrative borders [8,33,35]. A junctional component with cells cytologically similar to those seen in the dermis and pagetoid upward migration of melanocytes may be present [8]. Although random cytologic atypia may be present in conventional DPN, the atypia found in atypical DPN appears more than random and is most conspicuous within the areas of hypercellularity [8,33]. Lesional cells demonstrate moderate to severe cytologic atypia, including pleomorphism, high nuclear to cytoplasmic ratios, conspicuous cherry-red nucleoli, variable cytoplasmic melanin pigment, amphophilic cytoplasm, rare multinucleation, and 1 to 3 mitotic figures per square millimeter [8,33,35]. Manca et al. included lesions with up to 5 mitotic figures per square millimeter within their cohort [33]. Magro et al. include the presence of marginal mitotic figures (i.e., mitotic figures within 250 microns of the lateral and deep margins) as an atypical feature seen in atypical DPN [8]. Despite the increased mitotic activity, atypical mitotic figures are usually not seen [33]. A lymphocytic infiltrate may be commonly seen [33,34]. The case reported by Abraham et al. showed focal necrosis [34]. Atypical DPNs may exist in their pure form or as part of a combined nevus [33].

Atypical DPNs may show nodal deposits on sentinel lymph node (SLN) biopsy (Figure 6).

### 3.3. Immunohistochemical Features

Atypical DPNs demonstrate nuclear and cytoplasmic β-catenin expression [33]. The Ki67 proliferative index may be increased [34]. Atypical DPNs may show reduced expression of 5-hydroxymethylcytosine compared to conventional DPN [36]. Atypical DPNs that have progressed to melanoma may not show PRAME expression [35]. Some typical immunohistochemical features of atypical DPN are highlighted in Figure 7.

### 3.4. Molecular Biology

Atypical DPNs demonstrate mutational profiles similar to those of conventional DPN. Next-generation sequencing (NGS) performed on 21 atypical DPNs by Manca et al. showed frequent mutations of β-catenin pathway genes (most commonly *CTNNB1* mutations and less commonly *APC* mutations) and MAPK pathway genes (*BRAF*, *HRAS*, and *MAP2K1*) [33]. *IDH* mutations were found in 33% of cases [33]. Their single case with nodal disease demonstrated alterations in the β-catenin pathway and mutations in *IDH1* and *NRAS* [33]. Comprehensive mutation analysis showed low genetic heterogeneity for the main gene pathways [33]. No significant associations were found between specific gene mutations and histomorphologic features [33].

### 3.5. Cytogenetic Findings

Atypical DPNs usually demonstrate negative cytogenetic profiles on FISH and CGH. In the review of 40 cases of atypical DPN by Magro et al., FISH targeting 6p25 (*RREB1*), 6q23 (*MYB*), 11q13 (*CCND1*), and Cep6 performed in 10 of the 40 cases showed cytogenetic abnormalities characteristic of melanoma in three cases despite borderline histomorphologic features [8]. Oligo-array-based CGH performed in six of their cases (including two of the cases with melanoma profiles on FISH) did not show any significant chromosomal abnormalities [8]. In a FISH analysis of 13 atypical DPNs performed by Muhlbauer et al., only 3 lesions demonstrated abnormal FISH profiles [3]. The case reported by Abraham et al. did not demonstrate cytogenetic abnormalities on CGH [34]. In the case reported by Isales et al., FISH targeting *RREB1*, *MYB*, *CCND1*, Cep6, 9p21 (*CDKN2A*), and Cep9 performed on the initial atypical DPN lesion was negative [35].

Atypical DPNs may demonstrate normal cytogenetic profiles on initial biopsies at the borderline stage but then demonstrate chromosomal aberrations once they have morphologically progressed to melanoma [3]. One of the cases in the series reported by Magro et al. demonstrated normal cytogenetic profiles on both FISH and CGH on the initial biopsy but demonstrated unequivocal histomorphologic and cytogenetic progression into a DPN-like melanoma on subsequent biopsies (gains of *RREB1* and *CCND1* and loss of *MYB1* on FISH and amplification of 3p21.1-p11.1 and loss of 8p23.3-p11.21 on CGH) [8]. In the case reported by Isales et al., FISH targeting *RREB1*, *MYB*, *CCND1*, Cep6, *CDKN2A*, and Cep9 performed on the initial atypical DPN lesion was negative [35]. However, when the lesion recurred and was thought to be more morphologically consistent with melanoma, FISH showed clonal gains at 6p25.35.

### 3.6. Prognosis and Treatment

Although SLN deposits may be common in atypical DPN, reported transformation to melanoma and fatal outcomes are rare.

In the review of seven patients with atypical DPN by Magro et al., four patients developed positive SLN, and one patient who developed recurrence as melanoma died of widespread metastatic disease [9]. In their larger series of 40 patients with atypical DPN, Magro et al. had follow-up data for 37 of the 40 patients, with the follow-up period ranging from 5 months to 5.42 years (mean, 2.22 years) [8]. Thirty-five of the patients underwent re-excision, most commonly wide re-excision with 1- to 2-cm margins [8]. Of the 19 patients who underwent SLN biopsy, 6 had small subcapsular tumor deposits. Of these six patients, one had extensive parenchymal disease and developed recurrence one year after the wide excision; four underwent completion lymphadenectomy but did not have additional nodal disease; and two received adjuvant interferon alpha therapy [8]. Thirty-four patients remained without further metastatic or recurrent disease [8]. Two patients who were not initially diagnosed as having atypical DPN and did not undergo wide re-excision developed DPN-like melanoma and widespread metastases within three years and four years, respectively, after initial diagnosis and subsequently died [8]. A patient with atypical DPN who did undergo re-excision and had a positive SLN but normal CGH findings later developed recurrence as melanoma [8]. Two of the three patients with atypical DPN and positive FISH findings underwent SLN biopsy, and only one was positive for lymph node deposits [8]. All three patients remained alive and well at the time of the publication [8].

In the case reported by Isales et al., the patient did not undergo re-excision, despite the presence of positive margins on the initial biopsy, and developed recurrence 5 years later as melanoma with widespread SLN disease, including extracapsular extension, and 1 of 24 lymph nodes positive on completion lymphadenectomy [35]. The patient was treated with pembrolizumab with no subsequent recurrence for four years [35].

In the series of 21 cases reported by Manca et al., most patients were treated with simple excision without a further surgical procedure. Two patients underwent re-excision; one of these patients also underwent a SLN biopsy, which showed a positive node, with subsequent complete lymphadenectomy demonstrating 1 of 20 lymph nodes positive [33]. Follow-up data for 19 patients with a mean follow-up period of 38.1 months (range: 5–226 months) showed that all patients were alive and well and demonstrated no other local recurrences or distant metastases [33].

The patient reported by Abraham et al. had multiple small nodal deposits on SLN biopsy [34].

As outlined, a wide variety of treatment approaches have been utilized in cases of atypical DPN, including no further treatment after initial biopsy, re-excision with conservative to wide margins, SLN biopsy, completion lymphadenectomy, and systemic therapy. The most cautious course of action in atypical DPN, like in other histologically ambiguous cases, may be to manage these lesion as melanoma, however, this may be influenced by key factors including the age of the patient and the grade of cytological and architectural atypia [2,8,37]. It has been recommended that treatment of atypical DPN include complete excision with wide clear margins up to 1 cm [8,37,38]. We also recommend close clinical follow-up, which may include occasional full body examination, although definitive clinical guidelines are not available on this [14]. Although discussion of SLN biopsy may also be considered, in their reprise of the atypical DPN case by Abraham et al., McCalmont and Bastian emphasized that SLN biopsy is controversial in the evaluation of ambiguous melanocytic lesions and should be used as a staging tool rather than a diagnostic or predictive one [39]. There may be false positives and SLN biopsy is not considered standard management.

## 4. DPN-like Melanoma

### 4.1. Clinical Features

DPN-like melanoma has also been termed plexiform melanoma [40]. DPN-like melanomas usually affect young patients and are present on the face or upper body [40]. Like most melanomas, clinically, these lesions may demonstrate asymmetry, poor circumscription, color variegation, large size, and evolution over time.

### 4.2. Histopathological Criteria

DPN-like melanoma has histomorphologic features reminiscent of conventional DPN [40] but may demonstrate an atypical junctional component with pagetoid upward migration of melanocytes, irregular epidermal hyperplasia or epidermal atrophy and consumption, infiltrative growth patterns or nodular growth at the base, increased depth, ulceration, necrosis, inflammatory reaction, perineural invasion, lymphovascular invasion, increased cellularity, significant cytologic atypia, prominent mitotic activity, and atypical mitotic figures [5,14,35,40]. Atypical cytological features include marked pleomorphism, prominent pseudo-nuclear inclusions, abundant pale cytoplasm, and an irregular pattern of dusty cytoplasmic melanization (i.e., pulverocytes) [8,13,40]. These features are found throughout the lesion and are nonrandom [13,40]. Histomorphologic features are highlighted in Figure 8.

### 4.3. Molecular Biology

DPN-like melanoma may share activation of the WNT pathway with DPN, suggesting that some DPNs can progress to melanoma. A biphenotypic epithelioid and plexiform melanoma with DPN-like features was shown to demonstrate *BRAF* and *PTEN* mutations in both components but a *CTNNB1* mutation only in the DPN-like areas [41]. NGS performed on five cases of DPN-like melanoma by Yeh et al. showed MAPK pathway activating mutations of *BRAF* or *NRAS* in all cases and activating mutations in the β-catenin pathway in three cases [22]. All cases showed additional oncogenic alterations, including *CDKN2A*, *TERT*, *TP53*, *ARID1A*, and *TET2* mutations, and genomic copy number aberrations [22]. Isales et al., in their case of atypical DPN progressing to melanoma, showed mutations of *CTNNB1*, *NRAS*, *IDH1*, *ERBB4*, *GRIN2A*, and *MECOM* in both tumors using NGS [35]. The melanoma additionally had mutations of *TERT*, *DNMT3A*, and *PRSS3* and imbalanced chromosomal copy number gains in *BRCA2*, *RET*, *FGFR1* (also referred to as *FLT2*), and *IGF2* [35].

Evidence from molecular studies to date indicates that *BRAF* or *MAP2K* mutation leads to a conventional nevus, and subsequent *CTNNB1* mutation results in the phenotypic switch to DPN. Finally, additional molecular oncogenic alterations, including alterations in *CDKN2A*, *TERT*, and other genes, cause DPN-like melanoma [22].

### 4.4. Cytogenetic Findings

DPN-like melanoma may show abnormal cytogenetic profiles on FISH and CGH.

### 4.5. Prognosis and Treatment

DPN-like melanomas are associated with metastatic potential and an aggressive disease course [40]. It has been proposed that these lesions may be less aggressive than nonplexiform variants of melanoma of the same thickness [40]. Complete excision with wide, clear margins and SLN biopsy is recommended.

## 5. DPN Histological Variants

### 5.1. Plexiform Spindle Cell Nevus (PLEXSCN)

PLEXSCN is considered a variant of DPN [1]. Clinically, PLEXSCN lesions are darkly pigmented raised lesions that present in young adults without sex predilection on the upper torso [6]. The main histological differences between PLEXSCN and DPN are that PLEXSCN demonstrates plexiform architecture instead of a wedge shape, a more superficial depth with localization along the neurovascular plexus, and greater amount of intervening collagen between fascicles of lesional cells. PLEXSCN lesions may have a junctional component and can be part of a combined nevus. PLEXSCN lesions are usually benign; however, complete excision with clear margins is recommended [42]. An atypical form of PLEXSCN with increased mitotic activity, hypercellularity, and cytologic atypia and potential for lymph node involvement has also been described [42,43].

### 5.2. Clonal/Inverted Type A Nevus

Clonal/inverted type A nevus (also known as melanocytic nevus with focal atypical epithelioid cell components) is also considered a variant of DPN [1,44]. It demonstrates clinical and histological features similar to those of DPN [44,45]. However, clonal nevus is a more superficial dermis-based lesion composed of only epithelioid melanocytes and may include a junctional component [44,45]. Like conventional DPNs, clonal/inverted type A nevi are usually benign lesions [7]. Complete excision is recommended [44].

## 6. Conclusions

The DPN spectrum of melanocytic lesions is a diagnostically challenging group, for which cytogenetic findings may not always confer useful information for classification and risk stratification. This highlights the importance of astute histopathological examination in the evaluation of these lesions. Although NGS may have utility in identifying additional driver mutations, such as *TERT* promotor mutations, which may help in differentiating atypical DPN from DPN-like melanoma, this molecular technique may not be available at all centers. Until NGS enters widespread clinical use, in ambiguous cases, a cautious approach to management including complete excision with wide, clear margins and close clinical follow-up. SLN biopsy is controversial in these lesions. 

## Figures and Tables

**Figure 1 biology-11-00460-f001:**
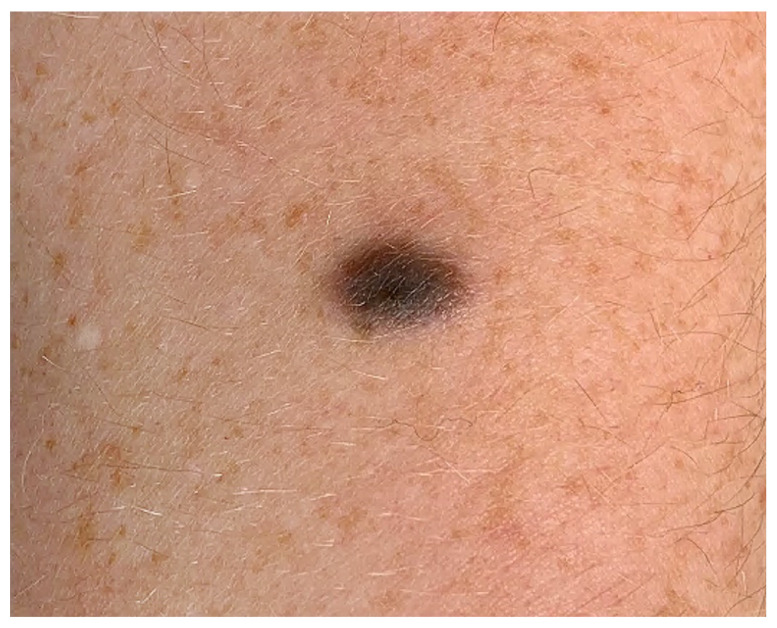
Deep penetrating nevi present as small, symmetrical, sharply circumscribed, pigmented papules or nodules.

**Figure 2 biology-11-00460-f002:**
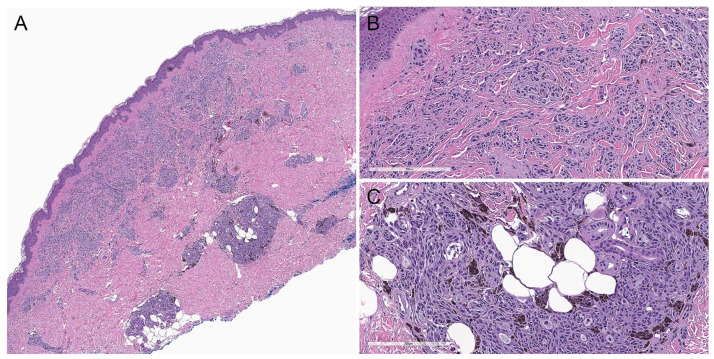
Conventional deep penetrating nevus. (**A**) There is a wedge-shaped proliferation of melanocytes in the dermis and subcutis with an overlying grenz zone (hematoxylin and eosin, 20×). (**B**) Lesional cells are arranged in fascicles, cords, and nests (hematoxylin and eosin, 100×). (**C**) The melanocytes are epithelioid and spindled, with amphophilic cytoplasm and admixed scattered pigmented melanophages (hematoxylin and eosin, 100×).

**Figure 3 biology-11-00460-f003:**
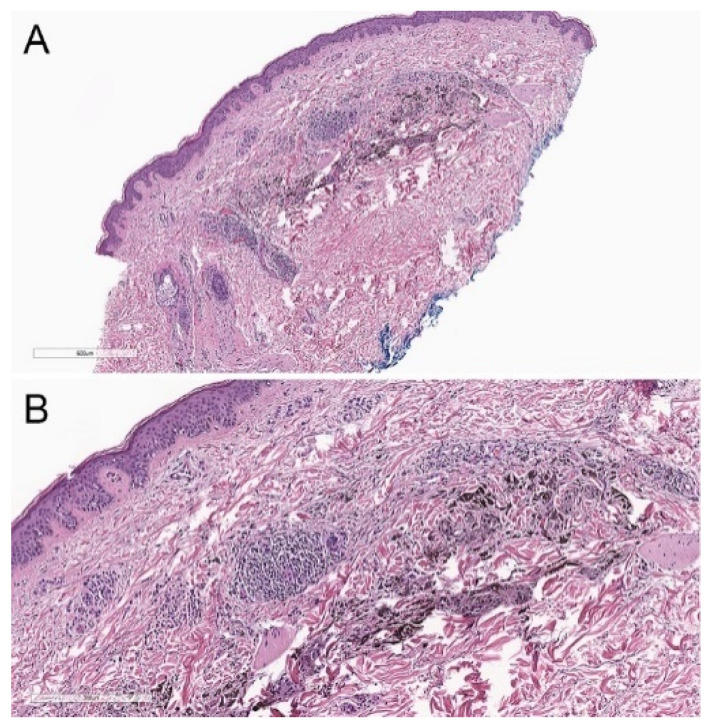
Deep penetrating nevus associated with a predominantly intradermal melanocytic nevus as part of a combined nevus. (**A**), Hematoxylin and eosin, 40×. (**B**) Hematoxylin and eosin, 100×.

**Figure 4 biology-11-00460-f004:**
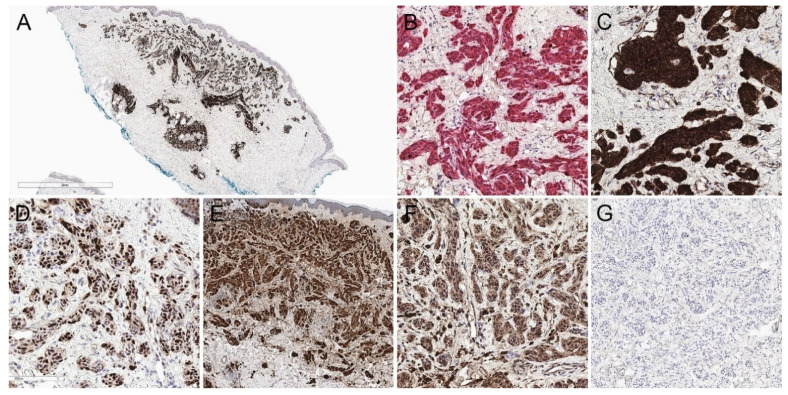
Immunohistochemical features of a deep penetrating nevus. (**A**) Diffuse expression of HMB45 is present (HMB45, 20×). (**B**) Dual MART1/Ki67 staining reveals a low proliferative index (MART1/Ki67, 200×). (**C**) Cytoplasmic, membranous, and nuclear expression of beta-catenin is present (beta catenin, 200×). (**D**) There is nuclear positivity for cyclin D1 (cyclin D1, 200×). (**E**) P16 is retained (p16, 40×). (**F**) BAP1 nuclear expression is retained (BAP1, 200×). (**G**) Lesional cells are negative for preferentially expressed antigen in melanoma (PRAME) (PRAME, 100×).

**Figure 5 biology-11-00460-f005:**
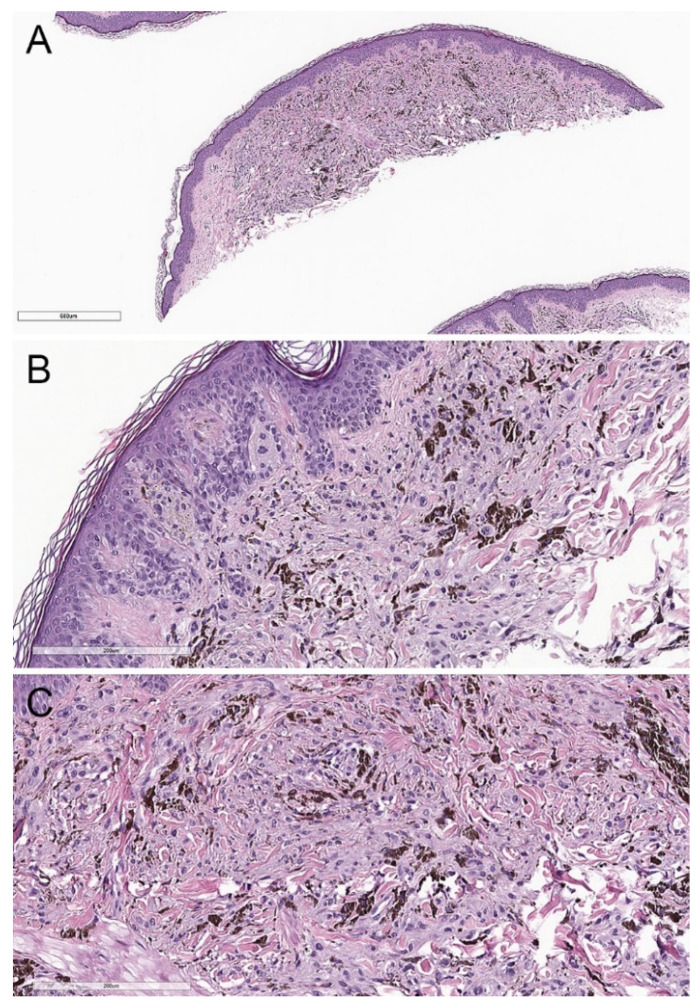
Atypical deep penetrating nevus. (**A**) The atypical architectural and cytological features exceed those seen in conventional deep penetrating nevi (hematoxylin and eosin, 40×). (**B**) The junctional component is predominantly nested and epithelioid with occasional pagetoid upward migration of atypical single cells (hematoxylin and eosin, 200×). (**C**) The dermal melanocytes are spindled and admixed with melanophages. Scattered cells show severe cytologic atypia (hematoxylin and eosin, 200×). In this case, findings on fluorescence in situ hybridization with probes for *RREB1*, *CCND1*, *MYC*, and *CDKN2A* were within normal limits.

**Figure 6 biology-11-00460-f006:**
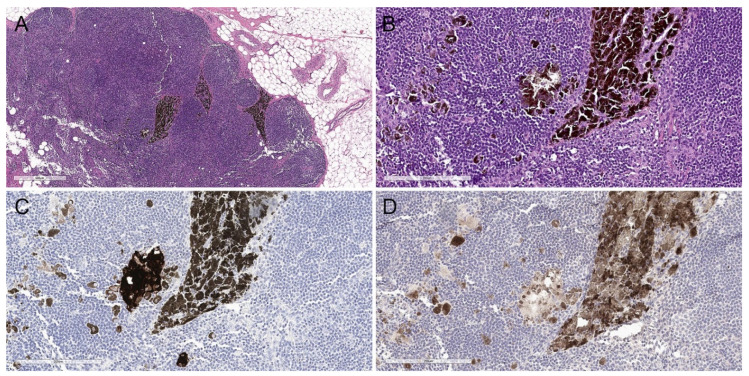
Sentinel lymph node deposits from a patient with an atypical deep penetrating nevus. (**A**) Hematoxylin and eosin, 40×. (**B**) Hematoxylin and eosin, 200×. Multiple clusters of coarsely and densely pigmented epithelioid cells are noted within the lymph node parenchyma and subcapsular space. (**C**,**D**) Lesional cells are highlighted on immunohistochemical studies with (**C**) a melanocytic cocktail (anti-MART1, HMB45, and anti-tyrosinase, 200×) and (**D**) SOX10 (SOX10, 200×).

**Figure 7 biology-11-00460-f007:**
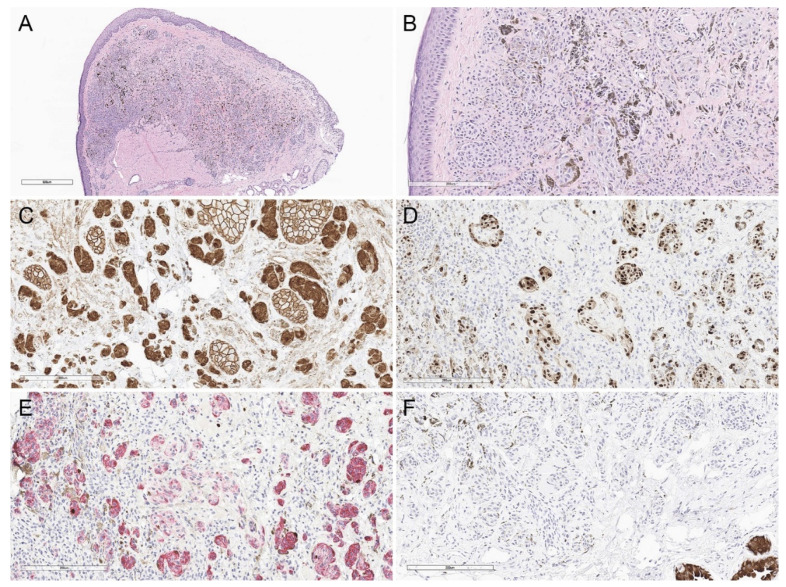
Atypical deep penetrating nevus from the eyelid margin. (**A**) Hematoxylin and eosin, 40×. (**B**) Hematoxylin and eosin, 200×. There is moderate cytologic atypia and rare mitotic figures. (**C**–**F**) Lesional cells demonstrate (**C**) membranous, cytoplasmic, and nuclear beta-catenin expression (beta-catenin, 200×), (**D**) nuclear cyclin D1 expression (cyclin D1, 200×), (**E**) a few proliferating melanocytes on a MART1/Ki67 immunostain (MART1/Ki67, 200×), and (**F**) no expression of preferentially expressed antigen in melanoma (PRAME) (PRAME, 200×).

**Figure 8 biology-11-00460-f008:**
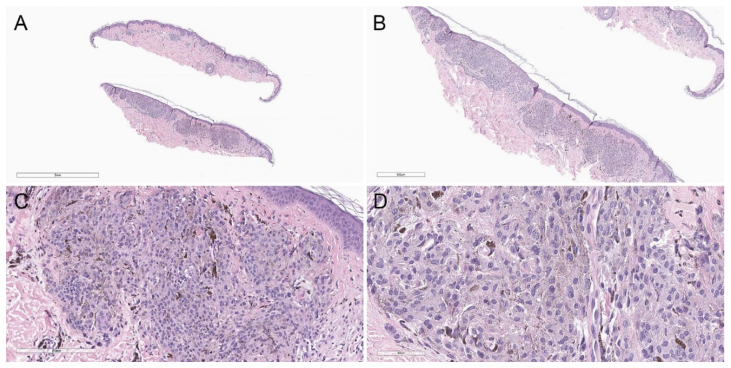
Deep penetrating nevus–like melanoma showing a compound, asymmetric melanocytic proliferation. (**A**) Hematoxylin and eosin, 20×. (**B**) Hematoxylin and eosin, 40×. (**C**) Lesional cells share cytomorphologic similarity with conventional deep penetrating nevus (hematoxylin and eosin, 200×). (**D**) However, there is increased cytologic atypia (hematoxylin and eosin, 400×). Additionally, immunohistochemical studies showed diffuse loss of p16, and the MyPath Myriad score (from qRT-PCR-based molecular analysis) was 2.0 (benign, −16 to −2; intermediate, −2 to 0; malignant, 0 to 10), supporting the interpretation of melanoma.

## Data Availability

Not applicable.

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
