# Peer review of "Conventional and Atypical Deep Penetrating Nevus, Deep Penetrating Nevus-like Melanoma, and Related Variants"

_biology, 2022, doi:10.3390/biology11030460_

Round 1

Reviewer 1 Report

The presented review is well organized and well written. My only suggestion is to include some information, if available,  on the dermoscopic features of  DPNs and its variants.

I believe that the present review, because of the specific subject could be addressed to a medical journal such as "Cancers" or maybe  "Cells".

Author Response

  • The presented review is well organized and well written. My only suggestion is to include some information, if available, on the dermoscopic features of  DPNs and its variants.

Thank you for your comments. Unfortunately the dermoscopic features of DPN are not well established, as we have outlined in section 2.1.

  • I believe that the present review, because of the specific subject could be addressed to a medical journal such as "Cancers" or maybe  "Cells".

This review article was written as an invited article for a special issue of Biology and hence that is why this journal was chosen.

Reviewer 2 Report

A very interesting narrative review about deep penetrating nevus and their more atypical and malignant variants; I found as a dermatologist the article very informative and eligible to be published after minor revisions:

  • Some other clinical images should be added, especially the ones of penetrating melanocytoma and more aggressive variants.
  • also, the dermoscopy of these lesions should be better defined.
  • In the introduction this paragraph "Melanocytic tumors are a heterogeneous group of lesions with wide variation in histomorphology and biologic behavior. Melanocytic tumors range from conventional acquired nevi, with unequivocally bland histopathological features and completely indolent
    clinical courses, to melanomas, with strikingly atypical histopathological features and inarguable potential for metastatic and aggressive disease" needs some referrals, such as doi: 10.3390/medicina57040359.

Thank You

Author Response

  • A very interesting narrative review about deep penetrating nevus and their more atypical and malignant variants; I found as a dermatologist the article very informative and eligible to be published after minor revisions.

Thank you for your comments.

  • Some other clinical images should be added, especially the ones of penetrating melanocytoma and more aggressive variants.

Unfortunately we do not have additional clinical photographs of these lesions.

  • Also, the dermoscopy of these lesions should be better defined.

Unfortunately the dermoscopic features of DPN are not well established, as we have outlined in section 2.1.

  • In the introduction this paragraph "Melanocytic tumors are a heterogeneous group of lesions with wide variation in histomorphology and biologic behavior. Melanocytic tumors range from conventional acquired nevi, with unequivocally bland histopathological features and completely indolent
    clinical courses, to melanomas, with strikingly atypical histopathological features and inarguable potential for metastatic and aggressive disease" needs some referrals, such as doi: 10.3390/medicina57040359.

An additional reference has been added.

Reviewer 3 Report

Dear Authors,

It was my pleasure to read your manuscript on deep penetrating nevus (DPN) and DPN-like lesions. Your review is a well-written and well-structured complex summary of the clinical, cytological and genetic features of these lesions with up-to-date and complete list of references. You use clear expressions and your descriptions are matched with numerous figures which makes the paper more appealing the the reader. The English is excellent.

My recommendation would be to accept the paper after minor revision.

Minor issues:

  1. In regard to atypical DPN (page 9) - WHO Classification of Skin Tumors introduce them as mostly benign lesions that rarely progress to melanoma (ref. 1). Nevertheless, you recommend to manage them as melanomas (ref. 2). Please consider mentioning the 2 key factors that should influence the management: the age of the patient and the grade of cytological and architectural atypia as mentioned by ref. 8 and ref. 40.

  2. Please provide how wide margins are recommended for atypical DPN and what to include in the clinical follow-up (palpation of the scar? regional lymph nodes? How long should be the follow up?).

    In my opinion it is worth to underline in a stronger way that SLNB has not been proved to have a prognostic value (many false positives) and should not be advised as a standard mananagement. Thus, you should not directly encourage to involve the patients in the decission process as I believe that after reading your paper almost all of them will be willing to undergo this invasive procedure, which will most likely bring any benefit to them, just like in atypical spitzoid tumours.

  3. Please provide the proposed management for clonal/inverted type A nevi, as you did for other lesions.
  4. If you introduce sentinel lymph node as SLN, and SLN biopsy as SLNB, please use it uniformely in the text.

Author Response

  • It was my pleasure to read your manuscript on deep penetrating nevus (DPN) and DPN-like lesions. Your review is a well-written and well-structured complex summary of the clinical, cytological and genetic features of these lesions with up-to-date and complete list of references. You use clear expressions and your descriptions are matched with numerous figures which makes the paper more appealing the the reader. The English is excellent. My recommendation would be to accept the paper after minor revision.

Thank you for your comments.

  • In regard to atypical DPN (page 9) - WHO Classification of Skin Tumors introduce them as mostly benign lesions that rarely progress to melanoma (ref. 1). Nevertheless, you recommend to manage them as melanomas (ref. 2). Please consider mentioning the 2 key factors that should influence the management: the age of the patient and the grade of cytological and architectural atypia as mentioned by ref. 8 and ref. 40.

This revision has been made.

  • Please provide how wide margins are recommended for atypical DPN and what to include in the clinical follow-up (palpation of the scar? regional lymph nodes? How long should be the follow up?).

This revision has been made. There are no clear guidelines on clinical follow-up.

  • In my opinion it is worth to underline in a stronger way that SLNB has not been proved to have a prognostic value (many false positives) and should not be advised as a standard mananagement. Thus, you should not directly encourage to involve the patients in the decission process as I believe that after reading your paper almost all of them will be willing to undergo this invasive procedure, which will most likely bring any benefit to them, just like in atypical spitzoid tumours.

We have added comments about the controversy of SLN biopsy as recommended.

  • Please provide the proposed management for clonal/inverted type A nevi, as you did for other lesions.

We have added a recommendation for complete excision per the literature.

  • If you introduce sentinel lymph node as SLN, and SLN biopsy as SLNB, please use it uniformely in the text.

This revision has been made.